# Saporin as a Commercial Reagent: Its Uses and Unexpected Impacts in the Biological Sciences—Tools from the Plant Kingdom

**DOI:** 10.3390/toxins14030184

**Published:** 2022-03-02

**Authors:** Leonardo R. Ancheta, Patrick A. Shramm, Raschel Bouajram, Denise Higgins, Douglas A. Lappi

**Affiliations:** Advanced Targeting Systems, Inc., Carlsbad, CA 92011, USA; leonardo@atsbio.com (L.R.A.); patrick@atsbio.com (P.A.S.); raschel@atsbio.com (R.B.)

**Keywords:** Saporin, immunotoxins, lesion, Alzheimer’s, pain, RIP, internalization, animal model, cancer, Alzheimer’s disease

## Abstract

Saporin is a ribosome-inactivating protein that can cause inhibition of protein synthesis and causes cell death when delivered inside a cell. Development of commercial Saporin results in a technology termed ‘molecular surgery’, with Saporin as the scalpel. Its low toxicity (it has no efficient method of cell entry) and sturdy structure make Saporin a safe and simple molecule for many purposes. The most popular applications use experimental molecules that deliver Saporin via an add-on targeting molecule. These add-ons come in several forms: peptides, protein ligands, antibodies, even DNA fragments that mimic cell-binding ligands. Cells that do not express the targeted cell surface marker will not be affected. This review will highlight some newer efforts and discuss significant and unexpected impacts on science that molecular surgery has yielded over the last almost four decades. There are remarkable changes in fields such as the Neurosciences with models for Alzheimer’s Disease and epilepsy, and game-changing effects in the study of pain and itch. Many other uses are also discussed to record the wide-reaching impact of Saporin in research and drug development.

## 1. Introduction

Ribosome-inactivating proteins (RIPs) have long been thought of as potentially perfect payloads for the treatment of human tumors; they could be tools to make Ehrlich’s Magic Bullet to remove harmful cells without harming healthy ones [1]. A common property shared among RIPs is their highly specific RNA N-glycosidase activity that cleaves the glycosidic bond of a single adenine base (A_4324_ in rat) from the ribosomal RNA of the large subunit 28 s [2]. RIPs are very potent, often active in the picomolar range. They can be targeted, as their corresponding bacterial toxins are, by attachment to molecules that bind to the cell surface and cause internalization. Some of the RIPs have targeting agents already attached and are per se toxins (e.g., ricin, abrin); some do not (e.g., Saporin, gelonin) and are as toxic as a cucumber, that is, not at all. In these latter cases, if the proper targeting agent is used, a specific antibody for instance, they can be very specific and very toxic to many different cell types that bind the targeting moiety, especially cancer cells.

That’s the upside. As RIPs had been utilized, some problems were found, and many of them have been resolved over the years. The presence of attached sugars created issues for pharmacokinetics because of binding to various membrane sugar receptors [3]. Sometimes the RIP was surprisingly inactive against the target cell type without artificial ‘enhancement’ [4]. Worst of all was the immunogenicity of the plant protein when injected systemically into a human. This also occurs with the bacterial toxins. Giansanti et al. provide a comprehensive discussion of the problem of immunogenicity of plant RIPs and discuss a comparative analysis of the research that has been done to remedy it. This same group compares the primary structure of Saporin to three other Type I RIPs that have been investigated regarding immunogenicity (bouganin, trichosanthin, and alpha-momorcharin). This examination shows that a YYF or YFF motif is located in all except bouganin, and a KR motif in trichosanthin is not conserved. Considering the conserved motifs, it was proposed that introducing a substitution at these points could positively affect the immunological epitopes without disrupting the catalytic activity [5]. As another approach to remedy the immunogenicity issue, Pastan’s laboratory explored a different route by identifying and eliminating the B cell epitopes via genetic means [6]. The immunogenicity was lowered through removal of the responsible amino acid sequences, through genetic means [6]. Despite successful work over several years, it appears pharmaceutical companies have adopted the idea of targeting very toxic small molecules. An example is calicheamicin, which cleaves double-stranded DNA, causing cell death [7], and is 4000 times more potent than doxorubicin. These smaller molecules are not immunogenic.

This article focuses on the use of one of these RIPs, Saporin. Research activity with Saporin is an area of avid interest. A cadre of researchers are taking the approach that the use of enhancers, especially the saponins or their derivatives from the same plant, *Saponaria officinalis*, will reduce the dosage needed for efficacy to a level that has a lesser immunogenicity. Others have had the idea that Saporin can be used fruitfully now as a research tool. It is some of those efforts that will be covered in this review. Bolshakov et al. [8], have well-covered some of this territory; we will refer to them in some cases, but amplify those areas they did not discuss.

One of the things that is reliable with Saporin is that if it is part of a complex that binds to a cell surface target, it will be internalized. This is not always the case with some of the toxin enzymatic chains. With Pseudomonas exotoxin, quite a bit of work went into the addition of sequence pieces outside of the enzymatic chain that were necessary for causing internalization and cell death [9,10,11,12,13]. This was accomplished primarily by the group of Ira Pastan over many years and the result is an anti-tumor drug, Lumoxiti. In addition to producing a cancer drug, the work of the Pastan group to diminish immunogenicity by the toxic moiety is a model for plant toxin workers.

There are, after all, a number of steps on the way: binding of the targeted toxin to the cell surface target, internalization, intracellular processing and trafficking to reach the cytosol, and inhibition of protein synthesis through ribosome-inactivation or one of the other players in protein synthesis, elongation factor 2 [14]. Saporin per se, and its related brother gelonin, are able to enter the cell along with its targeting agent with no extra additions to their sequences.

### 1.1. RIPs in Cancer Research

Saporin was first characterized from *Saponaria officinalis* by a group at the University of Bologna under the direction of Fiorenzo Stirpe [15]. It was identified as a good-sized chromatography peak of the soluble extract of the plant and was termed peak six, or SO-6, with high activity and high stability. Stirpe collaborated with Phillip Thorpe’s group in England to publish activity of a Saporin immunotoxin in a lymphoma model [16]. In a telling comment, the authors stated that, “Ricin A-chain coupled to OX7 antibody was one hundredfold to one thousandfold less effective than OX7-Saporin as an antitumor agent in vivo”. Over the years, a Saporin-targeted toxin has been used many times for the treatment of human disease [17,18,19], but has not yet been approved as an anti-cancer drug component. Efforts to produce an approved Saporin as a cancer drug continue, as seen recently [5] and in other articles contained in this issue of Toxins. Saporin has had difficulties in becoming the Magic Bullet for cancer treatment despite a significant amount of work. It is very helpful to recognize the excellent review by Polito et al. and the very active group at the University of Bologna [20]; lessons from this review published over 10 years ago are still relevant today. They cite the formation of antibodies against Saporin [21], vascular leak syndrome [5], and hepatotoxicity [22]. The first two may be remedied by genetic manipulation of Saporin [5,23]. The third could be rendered less problematic in clinical trial.

It is appropriate to mention that systemic administration of Saporin conjugates would dictate nearly immediate contact with tumors from systemic sources, as opposed to solid tumors. In addition, solid tumors, by definition, have tumor cells that are not immediately accessible; they are hidden in the three-dimensional structure of the tumor. This is one reason that much of the experimental success with Saporin conjugates is in cells in the systemic fluid. As to the development of a Saporin conjugate as a cancer drug, the immunogenicity of Saporin has left cancer drug development companies looking to non-immunogenic payloads.

It is important to point out that Saporin research has evolved from isolating the protein directly from the plant into recombinant expression. This is demonstrated in the cancer research area, where a chimeric fusion protein of the amino-terminal fragment (ATF) of the urokinase-type plasminogen activator (uPA) and Saporin create highly active therapeutic compounds [24]. In contrast, Zarovni et al. demonstrated how a SAP-encoding plasmid construct (SAP-KQ), that was cleverly produced by site-directed mutagenesis on the active site, results in Saporin having reduced catalytic activity [25]. This work with Saporin as an alternative to the currently used suicide genes in cancer therapy has helped pave a way for future work, e.g., suicide nanoplasmids coding for RIPs [26]. This work describes how the issue of vector size for conventional expression plasmids and the production cost of minicircle DNA can be remedied via nanoplasmids.

Other issues for Saporin in cancer applications are: potency, non-target activity in vivo, controlling stoichiometry, tumor penetration, and manufacturing costs.

Potency. In over 27 years of experience producing Saporin conjugates, our laboratory has seen many conjugates that work impressively well, and some that do not. The failings are most often caused by the targeting agent, so it is best to move on to test other targeting agents whether it be another antibody, a ligand, a DNA aptamer, any material that better “sees” the membrane molecule that is the target.Non-target activity in vivo. If this occurs in development, it is time to drop the project.Controlling stoichiometry. This can be accomplished in one of two ways: DNA construction of a fusion protein, or careful determination and description of the chemistry process for the construction of the conjugation.Tumor penetration. The use of saponins appears to be a promising method for increase in penetration and resulting potency [27].Manufacturing costs. GMP manufacturing of Saporin conjugates has a high price tag with very few facilities available with the appropriate equipment to produce grams of drug.

### 1.2. RIPs in Neuroscience Research

The use of targeted toxins in the nervous systems has advantages over cancer systems. For one, the nervous system doesn’t replicate such as tumors do. Trying to target tumor cells that are constantly expanding and spreading to other areas of the body is quite difficult; reaching all these cells requires systemic delivery which can result in an immune response. In the nervous systems, binding by the targeting molecule—whether it is a ligand, antibody, aptamer, or lectin—will cause internalization in nearly all cases, allowing Saporin to enter the cell. Once inside the cell, Saporin will inhibit protein syn-thesis and the cell will die. Neighboring, non-target bearing cells will remain. Unlike targeting individual tumor cells, the depletion of one neuronal cell type will also affect its pathway. For example, with the use of SP-SAP (Substance P-Saporin), the elimination of the perception of chronic pain is accomplished with a single dose delivered, thereby not causing an immune response (see Section 5. Clinical Trial for Cancer Pain). Another advantage to targeting nervous systems is that once cells are removed by a targeted toxin, they are usually not replaced. This allows possibilities for experimental studies on the various cell types with removal and examination of the effect. Drug development has mixed results. Alzheimer’s Disease became a major source of money for both research and drug development. The field has become stuck in the same place, with few drugs for the treatment of the disease. Another disease, epilepsy, has gone in a diverse direction, with at least 20 drugs approved for treatment, but none of them being a flat-out cure, such as those one can think of in cancer treatment. It, too, is an avid area of research with large patient populations. Saporin has become an important tool in many areas of neuroscience research.

The first use of a Saporin toxin in the nervous system was presented at Dr. Arthur Frankel’s Second International Symposium on Immunotoxins in June of 1991. The use of fibroblast growth factor-Saporin (FGF-SAP) in the inhibition of restenosis after angioplasty was an exciting result, but the organizer of the session, Ronald G. Wiley, was more interested in a brief account of injection of FGF-SAP into the left corpus callosum and the effect on the opposite side. After the session, Wiley proposed the use of Saporin conjugated to an antibody to the low affinity neurotrophin receptor, which is selectively expressed on neurons of the rat cholinergic neurons of the basal forebrain. This resulted in the removal of these neurons and the following publication was greeted warmly by the neuroscience community that had long desired a specific toxin for those cells to make a model for Alzheimer’s Disease (AD) [28]. The resulting conjugates, described in the next section of this review, have been described as giving one of the best models for AD [29,30].

The second major landmark of the use of Saporin that occurred, again suggested by Wiley, was conjugation of Saporin to the peptide substance P. A fusion protein had been constructed by Fisher et al. [31]. This molecule showed promise, but the amino terminus of the peptide had to be converted to an amide for the peptide to bind to the receptor. That required use of an enzyme that had been discontinued by its distributor. Instead, the chemical conjugation of the peptide with a conserved amide was found to be extremely potent for cells expressing the substance P receptor [32]. Mantyh and colleagues used this conjugate to cause the ablation of pathological pain in several models [33,34]. The results were a major surprise to the pain field; as eminent pain researcher, Howard Fields commented at the time: “No one expected a result like this”. The material showed activity in a clinical trial of cancer pain. It also stimulated the field of pain and itch and helped to determine differences between them [35,36,37].

One of the reasons that we went off into different fields from cancer was the concern in the early days, by those in the cancer field, that targeted toxins were too potent, and could have a non-specific impact on non-target cells. This was seen a couple of times in cancer studies in which deaths occurred, usually associated with exactly that problem: the antibodies used in these studies also targeted normal cells and tissues, at low levels, but sufficient enough to wreak havoc.

The use of Saporin conjugates in the research world caused similar concerns, and authors submitting publications were required to carefully document specificity. They did; targeted toxins that were targeted properly showed tremendous, well-documented specificity [38,39,40]. Hundreds of articles gave reason to believe that this system was free of artifacts as other methods exhibited them. A search of PubMed for ‘Saporin’ and ‘artefact’ yields no result, while a search for ‘optogenetic’ and ‘artefact’ yields 68 hits.

Finally, I wish to point out another result of the commercialization of Saporin conjugates. Obviously, to the researcher, the ability to buy a very active conjugate ‘off the shelf’ is pleasing. A major reason is the lack of necessity of working through a material transfer agreement. Once the conjugate is bought by a researcher, it belongs to that person. There is no need to be concerned about intellectual property of the inventor to be transferred to an outside entity; the inventor, including her institute, retain the intellectual property. This has proved to be an important value addition.

Non-targeted Saporin itself is a popular reagent for many researchers. These scientists work on delivery methods and oftentimes use Saporin to detect internalization. The read-out is simple; if their method causes internalization, the cell will die. This is very simple to quantitate. Some examples of targeting methods are listed:Combined apoptotic body analogues for efficient targeted therapy [41]Targeted and intracellular delivery of protein therapeutics by a boronated polymer [42]Macropinocytosis-inducible extracellular vesicles modified with antimicrobial protein CAP18-derived cell-penetrating peptides for efficient intracellular delivery [43]Nanobody-targeted polymeric nanoparticles [44]pH-sensitive coiled-coil peptide-cross-linked hyaluronic acid nanogels [45]CD22 ligands on a natural N-glycan scaffold [46]Vectorization of biomacromolecules into cells using extracellular vesicles with enhanced internalization [47]Soluble T-cell receptors produced in human cells for targeted delivery [48]

While these papers describe something similar to rocket science, the use of Saporin is less technical. Purified, highly-active Saporin is produced by advanced targeting systems and can be used by these scientists to demonstrate the efficacy of their inventions.

I have asked my colleagues to write about areas in which Saporin has made an impact. These people deal daily with the customers that have questions; they understand what the value of this technique is to scientists and if there are questions, resolving those questions. They have chosen topics that they appreciate to provide important information about the use of Saporin conjugates in science in these days.

## 2. Targeting Alzheimer’s Disease

The World Health Organization estimates that around 50 million people have dementia worldwide, with around 10 million new cases every year. In 2015, the social and economic impact had an estimate of $818 billion for medical and social care costs, which equated to 1.1% of the global gross domestic product [49,50].

Alzheimer’s disease (AD) is the most common form of dementia and has many symptoms that often present themselves gradually. The most talked about are loss of memory, confusion, repetitiveness, and mood and personality changes, and loss of initiative. Signs of severe AD include seizures, increased sleeping and difficulty swallowing, which often leads to aspiration pneumonia, and inevitable death [49]. Even though AD was first described in 1906 [51] there has been no cure for Alzheimer’s yet, which makes the characterization and development of treatments a high priority.

Over the years, several animal models of AD have been established to reproduce behavioral changes mimicking those seen in AD. Different types of lesions were produced by employing electrolytes, excitotoxins, or other alkaloid substances that led to non-specific and incomplete killing in several brain areas [8,52]. Cognitive dysfunction was also induced by alcohol injection [53] and β-amyloid peptides, which led to inflammation and cholinergic hypofunction [54]. Verkhratsky et al. stated that when compared to those lesions, the injection of 192-IgG-SAP results in depletion of specific brain areas relevant to AD, which makes it a reliable tool to investigate and understand the etiology of AD and its effect on cognitive performance [55].

### 2.1. 192-IgG-SAP

192-IgG-SAP uses the mouse monoclonal antibody 192-IgG, which recognizes the rat low-affinity neurotrophin receptor (p75). This is conjugated to the ribosome-inactivating protein, Saporin. This anti-neuronal immunotoxin permanently eliminates cholinergic neurons of the basal forebrain (CBF), medial septum, and diagonal band of Broca, which provide critical inputs to the hippocampus [56]. 192-IgG-SAP also targets neurons of the nucleus basalis of Meynert, and Purkinje neurons of the cerebellum, but not projections to the amygdala. The majority of the cholinergic neurons that project to the amygdala do not express p75 and are therefore not affected by 192-IgG-SAP [57,58]. These choline acetyltransferase (ChAT)-positive/p75-negative neurons can be eliminated with a different Saporin antibody conjugate called Anti-ChAT-SAP. It is important to emphasize that the amount of cell death due to targeted Saporin toxins directly correlates with the number of cells that express the p75 receptor. When the clean depletion of cholinergic neurons by 192-IgG-SAP was first documented, many leading cholinergic experts such as Bigl [59], Baxter [40], Johnson [60], Thal [61], Buzsaki [62], Sarter [63], Mesulam and Geula [64], Dunnett [65], and Levey [66] have immediately and successfully used the material for their research.

192-IgG-SAP was first used for neuronal depletion in 1991 [28]. Four years later the first AD rat model was developed by intraventricular injection of the immunotoxin [60]. 192-IgG-SAP is used to create animal models for the study of behavior, neuronal loss, plasticity of other systems in response to experimental loss, replacement therapy, drug effects, and drug dependence. The CBF losses are linked to decreased neurogenesis in the hippocampus, a region involved in learning and memory [67], which correlates to loss of cholinergic projections from the medial septum area to the hippocampus that are significantly reduced in AD patients, which explains the memory loss observed.

Accumulating studies have reported deficient performance in learning, memory, and attention after intracerebroventricular (ICV) injection of 192-IgG-SAP. ICV injection of this immunotoxin leads to nearly complete depletion of neocortical and hippocampal cholinergic neurons [68]. Wrenn & Wiley report that the larger the lesion extent, the larger the working memory impairment [40,58]. They also report that the outcome of neonatal ICV injection of the immunotoxin results in behavioral models of learning and memory only if the CBF lesion is widespread [58]. All the collected data from behavioral studies of rats with 192-IgG-SAP ventricular system injections illustrate that the p75-positive CBF neurons impact cognitive function.

The effect of 192-IgG-SAP on rodent brains can be seen in Figure 1 [28], Wiley, Oeltmann and Lappi showed more than a 99% loss of ChAT and nerve growth factor receptor (NGFR)-positively stained cells in the medial septum and nucleus of the diagonal band of Broca (Figure 1D,F) and decreased numbers of neurons in the CBF region (Figure 1B).

In the last 30 years, 192-IgG-SAP has helped many scientists in their AD research and in finding potential therapeutics for the deadly disease. Before 1995, when Steckler et al. addressed: “The role of serotonergic–cholinergic interactions in the mediation of cognitive behavior” [69], some researchers had already dealt with serotonergic–cholinergic interactions, but it was uncertain if neurotransmitter systems affect cognition directly or if the combination of serotonergic and cholinergic systems have a combined effect on behavior. Steckler’s research didn’t result in clear conclusions regarding this interaction, which led to another publication in 2002 to address the same question: whether other neurotransmitter systems also participate in behavioral capabilities commonly attributed to cholinergic functions [70]. 192-IgG-SAP lesions of CBF neurons were combined with 5,7-Dihydroxytryptamine (5,7-DHT) lesions of the ascending serotonergic projection system. The results showed that serotonergic depletion contributes to reduce the enormity of deficits in spatial working memory due to cholinergic damage. Interestingly enough, in 2006 it was implied that the “serotonergic system in the frontal cortex can compensate for diminished cholinergic function and support the investigation of the serotonergic system as a therapeutic target to treat AD” [71]. Since then, it has been established that serotonin plays a pathophysiological role in AD and that increase of serotonin neurotransmission is “effective in treating AD-related cognitive and behavioral deficits” [72].

The 192-IgG-SAP AD rat model has also been used to study therapeutic effects of human placenta-derived mesenchymal stem cells (pMSCs). Comparison between ICV or intravenous (IV) injection of these cells showed that both administrations led to cognitive recovery when compared to AD model animals. Cho et al. reported that “Acetylcholinesterase activity was significantly rescued in the hippocampus of rats injected with pMSCs post-lesion. Choline acetyltransferase did not co-localize with pMSCs, showing that pMSCs did not directly differentiate into cholinergic cells. The number of microglial cells increased in lesioned rats and significantly decreased back to normal levels with pMSC injection”. These results show that the administration of pMSCs leads to recovery of cholinergic neuronal populations and cognitive behavior [73].

Since neurogenesis is severely decreased in AD, it is of interest to see how adult hippocampal neurogenesis (AHN) can be restored. It is known that focused ultrasound (FUS) mediated blood-brain barrier opening increases neurogenesis, and a recent study was conducted to investigate whether FUS in a cholinergic degeneration rat model would increase adult hippocampal neurogenesis. It was concluded that FUS increased AHN and ameliorated spatial memory loss. A possible reason for this improvement is the upregulation of certain proteins in the hippocampus, which is a key factor for regulating AHN, synaptic plasticity, and neuroprotection [74].

In addition, the 192-IgG-SAP rat model created by ICV administration was used to show that deep brain stimulation (DBS) can be used as a treatment for dementia. Jeong et al. performed an experiment with four groups of rats that included an un-lesioned control, cholinergic lesion, cholinergic lesion, plus medial septum (MS) electrode implantation, and cholinergic lesion plus MS electrode implantation and stimulation. A water maze experiment was used to test spatial memory and showed that the lesion groups’ performances were slower when compared to the stimulation group, which performed as well as the control group. Lesion and implantation groups had a decrease in acetylcholinesterase activity in the hippocampus when compared to the stimulation and control group. The hippocampal neurogenesis was even increased in the stimulation group. These “results revealed that DBS of MS restores spatial memory after damage to cholinergic neurons” [75].

Another important task is to identify and understand pathways involved in AD that can help in discovering and developing treatments and/or prevention of the disease [76]. Research has shown that neuroinflammation plays a significant role in the pathogenetic process of AD. An especially important role belongs to microglia, which represent the main immune cells of the brain, as they protect and regulate brain development, neuronal activity, and survival by ingesting pathogens, including amyloid beta (Aβ), and damaged neurons in response to bacterial and viral infections [77]. In AD, microglia bind to Aβ oligomers and fibrils, which results in inflammation, cytokine production, memory impairment, and intracellular amyloid accumulation [78]. The hypothesis that soluble, ligand-like Aβ oligomers are responsible for brain damage that leads to AD was first introduced in 1998 [79]. Research such as this is an example of one of the popular uses of the model, which is, among other things, to ameliorate the effects of the loss of the missing p75+ neurons by addition of manipulated cells. There are many other AD-related research subjects such as hormones, neurovascular units, or biomarkers that have also been studied.

### 2.2. 192-IgG-SAP Species Specific Alternatives

There are two conjugates in addition to 192-IgG-SAP that specifically kill p75 neurotrophin receptor-expressing cells. ME20.4-SAP kills p75NTR-positive neurons in primate [80], rabbit [81], and sheep [82]. In mice, mu p75-SAP eliminates these neurons [83]. These additional immunotoxins allow for a wider scope, as research can occur in different species.

## 3. Behavior, Disease and Animal Models

Conjugates of Saporin have been used to create several models that are helpful in academic and pharmaceutical research (Figure 2).

### 3.1. Narcolepsy/Insomnia

Orexin-B-SAP, a conjugate of the Orexin-B (Hypocretin-2) peptide and Saporin, has been used extensively to study sleep and sleep disorders [84]. The generation of this tool began when researchers had determined the classic Doberman demonstration of narcolepsy was probably related to the human malady and perhaps genetically closely related. In 1999, these molecules had only recently been recognized by the Sakurai group at the University of Texas Southwestern Medical Center [85] and the group of J.G. Sutcliffe at the Scripps Research Clinic [86], and were shown to be involved in feeding behaviors. Since eating and sleeping are two of the most important behaviors that humans do and even like to do, interest in this newly characterized biological system was great, and the discovery of a sleeping aspect was widely reported in the press.

In 1999, Emmanuel Mignot at Stanford published an article showing that a mutation in the hypocretin/orexin receptor 2 caused narcolepsy in dogs [87]. An interesting result was demonstration of the difference between the cause of the human and dog behaviors: humans did not make hypocretin-2/orexin B and dogs had lost the receptor. In both cases, the axis was broken, and the similar behaviors were associated biochemically.

Due to the exciting and new findings in this system, Peter Shiromani published on how lateral hypothalamic lesions using the Orexin-B-SAP conjugate produce a narcoleptic-like sleep behavior in rats [88]. The creation of this model of narcolepsy has resulted in multiple studies that aim to find potential therapeutics for sleep disorders [89,90]. In addition to narcolepsy, insomnia occurs when Orexin-B-SAP is employed in other areas of the brain, such as the substantia nigra and ventrolateral preoptic nucleus [91,92].

### 3.2. Amyotrophic Lateral Sclerosis

Amyotrophic lateral sclerosis (ALS), also called Lou Gehrig’s disease, is centered around motor neuron deterioration. When these neurons die, muscles can’t function. Cholera toxin subunit B conjugated to Saporin (CTB-SAP) has a specificity towards motor neurons via GM1 ganglioside and has been used in different varieties to mimic complications in ALS. For example, tongue weakness and dysphagia are major complications in ALS patients and a model using CTB-SAP has been employed to induce hypoglossal motor neuron death to mimic the swallowing difficulties [93]. The authors made a point to express the importance and value of being able to study the impact of only this problem and to not have the other complications that are present in the other ALS models. Another example of a major complication in ALS is issues with the respiratory system, such as intermittent hypoxia. Intrapleural injections of CTB-SAP have been used to mimic this aspect of ALS and to study the impact of respiratory motor neuron death [93]. This targeted toxin has granted researchers the ability to study symptoms of ALS and motor neuron death and, therefore, we have a better approach at specific therapeutics. A group, led by Dale Sengelaub, was able to determine that testosterone prevents motor neuron atrophy through the model created by injecting CTB-SAP [94,95].

Gulino and Leanza have collaborated on work into neuroplasticity in the event of spinal motoneuronal loss, in a model utilizing CTB-SAP for depletion of spinal motoneuron populations in the gastrocnemius muscle. This is a simple and straightforward method to investigate the possible occurrence of compensatory changes in both the muscle and spinal cord. The results showed that, following the lesion, the skeletal muscle became atrophic and displayed electromyographic activity similar to that observed in ALS patients [96,97,98,99,100]. CTB-SAP is a prime example of how a single Saporin conjugate can be used in a variety of ways. Various areas of injection and the corresponding cells that are lesioned are outlined in Table 1.

### 3.3. Parkinson’s Disease

One of the main ways to research Parkinson’s Disease (PD) is to use the neurotoxin 6-hydroxydopamine (6-OHDA), which selectively destroys dopaminergic and noradrenergic neurons in the brain and can induce Parkinsonism in laboratory animals. Although a valuable tool, there are issues with using this model such as not producing extra-nigral pathology or Lewy body-like inclusions [111]. A Saporin conjugate, such as Anti-DAT-SAP, that utilizes a monoclonal antibody to the dopamine transporter conjugated to Saporin, can also lesion dopaminergic neurons [112]. It’s important to note the that the lesioning becomes more specific than 6-OHDA because the lesioning only occurs in places that express the dopamine transporter. Saporin conjugates are as specific as the targeting moiety of the conjugate and therefore there are constantly new targeting agents being attached to Saporin in order to create different, unique, robust, or better targeted toxins.

### 3.4. Itch

Various Saporin conjugates have been vital in discovering the mechanism behind itch. Understanding the pathway that itch is correlated to is important because one in five people experience chronic itch at least once in their life [113]. Chronic itch is also poorly treated, and treatments are often ineffective. The question that was asked to help deepen our knowledge on itch was—do itch and pain traverse the same pathways? The answer—no. Through selective ablation of neurons, itch and pain pathways have become better understood. For example, Bombesin-SAP and Substance P-SAP (SP-SAP) reduce itch in ovalbumin-sensitized mice [35]. Nppb-SAP helped clarify the circuitry for itch response in mice and that Natriuretic Peptide B (Nppb) is an itch-selective neuropeptide [114]. These advances are uncovering mechanisms of how itch is relayed to the brain, which helps identify targets for improved treatments that reduce or block abnormal itch [36].

### 3.5. Epilepsy

One of the animal models developed to produce temporal lobe epilepsy in rats has been via systemic injection of pilocarpine. This model, developed by the Turski group in 1983 [115], has certainly been a valuable tool in the field of researching epilepsy. However, the mortality rates of rats are around 30–40% when injecting with the necessary amount of pilocarpine to induce the syndrome [116]. The high mortality rate of this model and the number of antiepileptic drugs (AEDs) that exist means that there is room for improvement of a model for temporal lobe epilepsy and treating the disease itself.

A newer model that creates hippocampal sclerosis and chronic epilepsy very similar to the occurrence in the human disease has been developed by injection of stable Substance P conjugated to Saporin (SSP-SAP) into the hippocampus [117]. This model eliminates the interneurons that suppress the excitatory neurons. Release of excitatory neurons causes spontaneous focal motor seizures and extreme hippocampal sclerosis. This opens the possibility to develop specific pharmaceutical agents with known definitive effects for epilepsy.

### 3.6. Gastroenterology

Saporin has had an impact in the study of inflammation and pain caused by disorders of the stomach, intestines, and pancreas. Here are just a few of the areas where scientists using Saporin conjugates have made important discoveries that could lead to treatments for these disorders:

Crohn’s Disease: an inflammatory bowel disease associated with several changes in the immune system, including an increased number of infiltrating macrophages that release a variety of cytokines that are responsible for inflammation.

Kanai et al. investigated the role of macrophages in a 2,4,6-trinitrobenzene sulfonic acid-induced colitis mouse model [118]. A 20 μg dose of a CD11b antibody conjugated to Saporin (Mac-1-SAP) was administered parenterally in the tail vein. Seven days after treatment, mice showed no evidence of intestinal inflammation.In this second study by the Kanai group, Mac-1-SAP was administered as a single intravenous (IV) injection that significantly reduced the amount of intestinal inflammation [119].The study by Yamazaki et al. used the same colitis mouse model referenced above and investigated the role of mucosal T cells that express high levels of interleukin-7 receptor (IL-7R) in the development and treatment of chronic colitis. A custom conjugate of an antibody to the IL-7R and Saporin was administered via intraperitoneal (IP) injection once a week for 6 weeks to 20 to 24-week-old mice. Selective elimination of IL-7R-expressing T cells ameliorated established, ongoing colitis [120].

Pancreatitis: inflammation of the pancreas.

A rat model of persistent experimental pancreatitis was used to determine the role of descending pathways in the pain caused by pancreatitis [121]. Rats received 1.5-picomolar injections of the targeted toxin Dermorphin–Saporin (a conjugate of the peptide dermorphin and Saporin; MOR-SAP) into each side of the rostral ventromedial medulla. Although the ablation of mu-opioid receptor-expressing neurons by MOR-SAP did not prevent the initial expression of pancreatitis pain, chronic pain was eliminated thereby linking the maintenance of pancreatitis pain to descending pathways. This treatment also prevented the increase of spinal dynorphin content.Macrophages perform different functions depending on the tissue type. The specific differentiation that macrophages undergo in response to their environment is called polarization. Criscimanna et al. used a mouse pancreatic lesion model to examine the polarization of macrophages into the two distinct states known, M1 and M2 [122]. Mice received 20 μg of Mac-1-SAP in a tail vein. The results of this study demonstrate that various aspects of macrophage polarization are required for pancreatic regeneration. The authors state that, “Additional study of these processes and signals might lead to new approaches for treating Type I diabetes or pancreatitis”.

Diabetic Encephalopathy: one of the severe microvascular complications that can occur in patients with diabetes. It causes inflammation in the brain which can lead to dementia and central nervous system impairment.

In a study conducted by Wang et al., diabetes was induced in mice by injection of streptozotocin (STZ). In order to investigate the role of macrophages in the development of diabetic encephalopathy, IP injections of Mac-1-SAP were administered twice weekly. Mice receiving Mac-1-SAP had greatly reduced numbers of inflammatory macrophages in the brain without affecting blood glucose, serum insulin, glucose responses or beta cell mass [123].

The role the brain plays in the gastrointestinal (GI) tract can be explored with Saporin conjugates. Work has been done in the nodose ganglia with a conjugate created with the hormone cholecystokinin (CCK-SAP).

CCK-SAP was used to induce neural lesioning of vagal afferent neurons while sparing vagal efferent neurons in the nodose ganglia of rats [124].In 2018, Suarez et al. injected CCK-SAP into the nodose ganglia to “eliminate ~80% of GI-derived vagal sensory input to the brain while leaving intact all brain-to-gut vagal motor signaling, and supradiaphragmatic and colonic vagal sensory signaling” [125]. This technique identified a previously unknown role for the gut–brain axis in memory control.

### 3.7. Noradrenergic Lesioning/Anti-DBH-SAP

The aforementioned 192-IgG-SAP was an incredibly useful new tool that helped propel the search for other conjugates. One of the next promising lesioning tools would be one that targeted the noradrenergic system. As stated previously, 6-OHDA ablates both dopaminergic and noradrenergic neurons; the ability to ablate only noradrenergic neurons came with the creation of Anti-DBH-SAP. Anti-DBH-SAP is a conjugate of a monoclonal antibody to the noradrenergic specific enzyme dopamine beta hydroxylase (DBH) and Saporin [38]. It is important to note that unlike most Saporin conjugates, Anti-DBH-SAP is targeted towards an enzyme that spends time on the cell membrane and is not targeted towards a cell surface receptor.

There are many groups that have worked with Anti-DBH-SAP. Some of the initial publications that have detailed the selectivity and extent of lesioning by Anti-DBH-SAP were done by Patrice Guyenet’s group, where they focused on the role of specific bulbospinal neurons [126,127,128]. A large part of their published research has been dealing with the C1 neurons located in the rostral ventrolateral medulla that deal with stress, pain, catecholamine release, and other noradrenergic functions. Many of the publications that utilize Anti-DBH-SAP are trying to determine mechanisms such as: receptor signaling [129], behavioral changes [130], and studying Alzheimer’s disease [131]. Anti-DBH-SAP is one conjugate that highlights the fact that the same tool can be used in various research areas, in unique ways, to create and answer thoughtful questions.

### 3.8. OX7-SAP

One of the first and more potent Saporin conjugates made commercially available to target and destroy neurons was OX7-SAP, a conjugate of the monoclonal antibody to rat Thy 1 and Saporin. This was the first anti-neuronal immunotoxin effective in vivo. The OX7 antibody was one of the original antibodies to cell-surface proteins, developed by England’s Medical Research Council (MRC), which was home to Crick, Perutz, Brenner, and other influential researchers. The conjugate targets all rat neurons and has been used in a variety of different experiments [132]. For example, Soucacos and colleagues used OX7-SAP as a lesioning agent to inhibit neuroma in continuity formation after controlled nerve damage [133]. This application is one of many examples of neuronal death by suicide transport. Due to the success of OX7-SAP and its activity in vivo, the first receptor-specific immunotoxin was created—192-IgG-SAP (see Section 2: Targeting Alzheimer’s Disease).

## 4. Secondary Conjugates and Streptavidin-ZAP

Immunotoxins made with Saporin have provided robust tools, especially in the fields of neuroscience [5,8,134] and cancer [5,135]. As studies in these fields continue to expand, finding a modular way of creating targeted toxins becomes more desirable. Not all laboratories are created equal and having the scientific background, time, and funds needed to produce these tools are often unavailable.

For the synthesis of an immunotoxin, choosing the right antibody is key. Particularly in the area of drug development, large numbers of antibodies need to be screened for suitability. In the year 2000, a tool was developed for evaluating antibody efficacy for use as an immunotoxin. The idea was first suggested by Till et al. [136] but was not commercially available.

A similar molecule, Mab-ZAP, was prepared at Advanced Targeting Systems (ATS) for commercial use [137]. Mab-ZAP consists of a polyclonal goat anti-mouse IgG antibody chemically conjugated to Saporin. ATS created the moniker ‘ZAP’ in place of Saporin for all its non-targeted secondary conjugates and has continued to use it in its commercial reagents. Its mechanism of action is simple: Mab-ZAP is mixed with a candidate antibody, the complex binds to a cell-surface antigen, and is internalized. Once internalized, Saporin inactivates the ribosomes, protein synthesis ceases, and this results in cell death. Mab-ZAP doesn’t have a way to enter a cell on its own and so only the cells expressing the specific antigen recognized by the primary antibody candidate is affected.

Considerable thought was put into the chemistry of the secondary conjugates and how they should be linked to Saporin. The exact methods have remained proprietary, but in general, a bi-functional, cleavable linker was chosen to be the most appropriate, providing an ~7 angstrom distance between Saporin and the secondary antibody. Type of linkage and location were important to address, especially regarding the antibody being able to still recognize and bind its target and Saporin retaining its cytotoxic effect. Because of the inherent characteristics of Saporin, such as its high isoelectric point, and resistance to proteases, just to name a few, purification was made easier, and highlighted why Saporin was a preferred payload over the various toxins already mentioned earlier.

Figure 3A–C depicts an overview of secondary conjugates constructed with either whole IgG or the fragment antigen-binding (Fab) region as the targeting agent. Mab-ZAP was the first secondary conjugate in an arsenal of tools that ultimately eliminate time-consuming and expensive steps of chemically linking each antibody candidate directly to Saporin.

Subsequently, more secondary conjugates were developed to expand the range of species. Initially, these tools utilized whole-molecule IgG secondary antibodies that recognize both heavy and light chains of primary antibodies. In theory, this presented a possible limitation if the bivalent nature of an antibody suggested that cross-linking could occur on the cell surface and contribute to a phenomenon known as ‘cap formation’. The ‘cap’ could potentially induce some level of endocytosis that would lead to cytotoxicity and produce a false positive for internalization of a primary antibody. The idea was that most cell membranes have a fluid mosaic structure, where the surface proteins are actually free to move or diffuse across the lipid matrix and randomly distribute within the cell surface [138]. The other side of that coin is that external agents can actually induce surface antigens to adopt nonrandom distribution. An example of this is on lymphocytes, where a multivalent antibody that can recognize and bind surface IgG will induce a cap formation. This redistribution of receptor sites is followed primarily by endocytosis and partly by shedding of the capped complexes, which leaves the cellular surface temporarily free of binding sites [138,139,140,141,142]. As a pre-emptive measure, new secondary conjugates were designed (given the moniker Fab-ZAP) using monovalent antibodies that would continue to recognize whole IgG, but lacked components that would contribute to capping (Figure 3B). Fab-ZAP uses a monovalent antibody, consisting of only the Fab region of IgG and which recognizes whole IgG, not discriminating between the fragment antigen-binding (Fab) or fragment crystallizable (Fc) region.

Figure 4 shows the data from a cytotoxicity assay using various secondary conjugates when compared to the direct conjugate, 192-IgG-SAP. Mab-ZAP + primary antibody complex produces a similar potency to the directly linked conjugate of Saporin to the same antibody and, more interestingly, Fab-ZAP and FabFc-ZAP + primary antibody produced a cytotoxic effect > 12-fold over the direct conjugate. Figure 4 also depicts an interesting phenomenon with secondary conjugates. It may be intuitive to think that using a higher dose of primary antibody induces a higher amount of cell death, but as seen in Figure 4, at the highest concentration of 192-IgG (10 nM = Log(−8)) there is a lessened amount of killing, at a 25-fold lower concentration when compared to the antibody. The explanation for this is that, at the higher concentrations of primary antibody, there are more unconjugated 192-IgG and fewer 192-IgG+Fab-ZAP complexes. The free 192-IgG then out-competes the conjugates for cell surface binding sites which, in turn, decreases the amount of Saporin being internalized, hence, less cell death.

With the popularity of the Fab-ZAP line of secondary conjugates, increased use was accompanied by new pitfalls. A drawback that had to be addressed was when using these secondary conjugates with B cells, e.g., those of non-Hodgkin’s lymphoma. Figure 5 depicts the endogenous expression of surface immunoglobulins (sIg) on Daudi cells (a human B lymphoblast) and the nonspecific targeting of secondary conjugates that recognize whole molecule IgG vs. an Fc-specific secondary conjugate. B cells express endogenous surface sIg and are anchored to the surface in both naïve and activated B cells by the Fc region of the molecule, while the Fab portion is exposed and can be targeted by a secondary antibody. To remedy this, building on the monovalent secondary antibody used to make Fab-ZAP, an Fc-specific version was created. Instead of recognizing whole molecule IgG, only the Fc region of the IgG heavy chain is targeted, allowing for confidence that cell death was antibody mediated.

The search for antibodies to use in therapeutic treatment of diseases continues to be ongoing, with the market of global therapeutic monoclonal antibodies valued at $115.2 billion in 2018 and expected to reach a revenue of $300 billion by the year 2025 [143]. With this continued interest, innovative tools will continue to be necessary to expedite the antibody screening process.

While all these secondary antibody conjugates fulfill the goal of a more modular and efficient way of creating of an immunotoxin, the issues of: (1) limitation to using an antibody as a targeting agent, and (2) instability in vivo, continued to be hindering factors. An answer to these issues came in the form of streptavidinylated Saporin (Streptavidin-ZAP).

Streptavidin is a tetrameric protein (molecular weight 53 kDa in its recombinant form), with each subunit able to bind a single biotin molecule. This highly stable interaction between biotin and streptavidin (rapid, essentially non-reversible, unaffected by most extremes of pH, organic solvents, and denaturing reagents) led to the development of a new wide-reaching tool for molecular surgery. By attaching streptavidin to Saporin, researchers who were looking to either screen new potential therapeutics or to study behavior after depletion of cells, could do so by harnessing the strongest known noncovalent biological interaction: the one between biotin and streptavidin (Ka = 10^15^ M^−1^). With this addition to the arsenal of secondary conjugates, Saporin conjugates were made available for use with not only different sections of an antibody (i.e., F(ab’)2; Fab, or scFv) but with anything that could be biotinylated, recognize a cell-surface antigen, and be internalized (Table 2).

## 5. Clinical Trial for Cancer Pain

### 5.1. The Road to Human Clinical Trials

In 1997, Patrick Mantyh published a paper in the journal *Science* entitled “Inhibition of hyperalgesia by ablation of lamina I spinal neurons expressing the substance P receptor” [33]. This publication made a huge impact in the field of pain and was followed by a series of television, radio, and news reports about this one-shot, targeted drug, a ‘magic bullet’, Substance P–Saporin (SP-SAP), that eliminates chronic pain while leaving acute pain intact (Figure 6). The 1997 publication was followed by a 1999 publication by Nichols et al. from the same laboratory entitled “Transmission of chronic nociception by spinal neurons expressing the substance P receptor” [34]. Following on the heels of the first publication which showed the specificity of Substance P-Saporin (SP-SAP) to a small subset of neurons in the spinal cord, this second publication in the journal *Science* attested to the fact that this inhibition of pain was likely permanent, as it still persisted after one year in the several rat models being used to study the drug.

Over the next nine years there were over 30 publications by independent researchers from around the world that tested SP-SAP in a variety of different applications. Then, in 2006, came the publication by Allen et al. from the Tony Yaksh laboratory at the University of California, San Diego, of a safety study and evaluation of intrathecal SP-SAP in dogs [144]. This began the journey to preparing a regulatory package that could be presented to the FDA to get approval for using SP-SAP in a clinical trial in humans. This was not a particularly easy task and took several years of communications with the FDA and collaborations with a number of different scientists who had specialties in fields of safety, toxicology, pain models, and regulatory processes. Nineteen more publications were seen in the next seven years, followed by another safety study in dogs published by Wiese et al. in the journal *Anesthesiology* [145].

### 5.2. Human Clinical Trial

In 2008, a pre-IND (Investigational New Drug) meeting was held with the United States Federal Drug Administration (FDA), and the Division of Anesthesia, Analgesia, and Rheumatology Products at the Center for Drug Evaluation (CDER). The meeting was positive and provided clear guidance for the next steps in the areas of chemistry, manufacturing, and controls (CMC), as well as the pre-clinical toxicology requirements for the IND. The CDER review panel indicated that the patient population was appropriate and noted that there was a clear unmet clinical need for drugs addressing a terminal cancer patients’ intractable pain. They determined that a terminal population was appropriate due to the irreversible elimination by SP-SAP of <5% of NK1R+ neurons in the spinal cord [144]. When the clinical trial began in 2014, it became evident that this population was also going to pose many problems in isolating and determining the efficacy and safety of SP-SAP in eliminating the perception of chronic pain. Terminal cancer patients receive an arsenal of medications that can have serious side effects (e.g., oxycodone, Keppra, Celebrex, Ambien, gabapentin, Dilaudid, etc.), on top of radiation and chemotherapy treatments.

In 2014, Wiese et al. described the FDA-required intrathecal safety study in dogs [145]. Later that year, the FDA gave the green light to begin a human clinical trial using SP-SAP in terminal cancer patients who are resistant to opioids such as morphine. The clinical trial in humans began at the University of Texas Southwestern Medical Center in Dallas under the direction of Dr. Carl Noe in the department of Anesthesiology and Pain Management and was sponsored by Dr. Arthur Frankel at UTSW’s Simmons Comprehensive Cancer Center. Dr. Frankel is a leading expert in the use of targeted toxins in humans. The primary endpoints of the trial were to monitor the effect of SP-SAP in patients by measuring, after a single intrathecal dose of SP-SAP: (1) the reduction of pain, and (2) the reduction of the frequency and amount of standard-of-care pain-relieving drugs and procedures. Two of the four patients treated clearly met these endpoints, with a reduction of pain medication by >20% during an 8 week period following treatment, with no evidence of toxicity, or neurological or cardiac abnormalities (Figure 7). The first clinical trial publication entitled “Preliminary results from a Phase I study of Substance P-Saporin in terminal cancer patients with intractable pain” was presented at the 2016 meeting of the American Association for Cancer Research [146,147].

Throughout the preclinical process, pharmaceutical companies were approached to find a partner more experienced and capable than Advanced Targeting Systems (a small business research supply company) was. Time and again, the response was negative. It became clear that the idea of a one shot pain relief treatment was not nearly as attractive as an oral medication that had to be taken several times a day—perhaps the rest of the patient’s life. Unable to partner or license with a drug development company, the clinical trial was put on hold in 2016 due to lack of sufficient funds to continue.

### 5.3. Veterinary Clinical Trial

In 2013, a veterinary clinical trial in dogs with bone cancer pain was begun by Dr. Dorothy ‘Dottie’ Brown at the University of Pennsylvania. This led to further publications and more information that would allow the filing of an application to get approval from the Center for Veterinary Medicine for treatment of dogs with bone cancer [146,148]. A minor use/minor species (MUMS) designation was granted for SP-SAP, providing extended market exclusivity to treat the >10,000 annual cases of canine bone cancer-related pain, and the ability to commercialize the drug as soon as conditional approval was given. One of the persuasive factors in this approval was a video of one the dogs that chronicled his condition before and after treatment with SP-SAP [149]. This study resulted in a publication in the journal *Anesthesiology* entitled “Intrathecal Substance P-Saporin in the dog: efficacy in bone cancer pain” [150]. An editorial comment by Ken-ichiro Hayashida on Dr. Brown’s publication discussed the aspects of the NK1 receptor (NK1R) targeting approach [148]. He stated that Substance P had long been known to be important in normal pain transmission but in the late 1990s, its receptor in the spinal cord (NK1R) began to be implicated in pathological pain. Substance P antagonists were tested in human trials but did not provide pain relief; this caused Hayashida to ask why then would elimination of NK1R-expressing spinal neurons work? He suggested that blockade of only one type of receptor is not sufficient because of all the other neurotransmitters of pain that still function on the pathology-producing neurons [148]. The elimination of NK1R+ neurons removes the pathology of chronic pain. This, however, does not affect the crucial perception of normal acute pain (e.g., hand on a hot stove, hitting your thumb with a hammer).

In 2016, after learning that it would cost over one million dollars to produce SP-SAP suitable for commercialization, the Center for Veterinary Medicine application was put on hold until funds could be raised to generate the drug and data to satisfy the FDA requirements. This was one of the painful learning processes a small business must endure in drug development. It did not occur to ATS that the drug being used in human clinical trials was not sufficiently validated to allow for approval and commercialization of a veterinary drug.

### 5.4. Preclinical Work Using SP-SAP

Extensive preclinical and clinical work has been done with SP-SAP. SP-SAP was commercially available to researchers from 1997–2007. During this time, it underwent the equivalent of beta testing, and it became clear, mainly from the researchers that had absolutely no connection with the inventors or developers of SP-SAP, that this material has profound inhibitory effects on pathologic pain. Researchers came to the following conclusions:SP-SAP in the upper cervical dorsal horn of adult rats reduced aversion to suprathreshold but not near-threshold levels of oral high intensity pain stimulant (capsaicin) [151].SP-SAP eliminated a pivotal component of the spinal circuits involved in triggering central sensitization and hyperalgesia [152].SP-SAP removed NK1R+ spinal projection neurons that project to higher brain areas and control spinal excitability—and therefore pain sensitivity—primarily through descending pathways from the brainstem. This implies central sensitization is stopped by SP-SAP [153].SP-SAP inhibited high level pain transmission in more complex pathologic pain models [154].SP-SAP attenuated the tactile and cold hypersensitivity and abnormal neuronal coding (including spontaneous activity, expansion of receptive field size) seen after spinal nerve ligation [155].Ablation of NK1R+ lamina I cells eliminates the ascending limb of a spinal–bulbospinal loop that engages descending facilitation [121].Substance P–Saporin decreased the ratio of NK1R+ neurons innervating the disc related to discogenic low back pain. Conclusion: SP-SAP may be a useful tool to investigate the mechanism of discogenic low back pain [156].The generation of intrinsic GABAergic transmission in the spinal cord appears dependent on NK1R+ neurons, yet despite the loss of GABAergic inhibitory controls after SP-SAP treatment, the net effect is a decrease in spinal cord excitability. Thus, activation of these cells predominantly drives facilitation (and elimination inhibits sensitization) [157].Since the same neuronal population drives descending facilitation and inhibition, the reduced excitability of lamina V/VI WDR (wide dynamic range) neurons seen after loss of NK1R+ neurons by SP-SAP indicates a dominant role of descending facilitation [158].Elimination of NK1R+ neurons with SP-SAP ends descending facilitatory pathways that function in central sensitization [159].In a dog safety study, data indicate no adverse toxicity at doses up to 10 times those necessary for producing loss of superficial NK1R+ neurons in a large animal model [144].

### 5.5. Preclinical Work Using Stable Substance P-Saporin (SSP-SAP)

SSP-SAP is a conjugation of Saporin and stable Substance P, the Sar9, Met(O2)11 analog of Substance P. The two amino acid replacements are at two sites of digestion by tissue proteases which allows this peptide to resist peptidase digestion, allowing a greater diffusion from the injection site before it is metabolized. It is considered a ‘new and improved’ version of SP-SAP for research purposes where the targeting and elimination of NK1R+ cells in a larger area is desired; it has not yet been studied sufficiently to progress into veterinary or human trials. SSP-SAP is commercially available, which has resulted in even more research discoveries when NK1R+ neurons are ablated (Table 3).

## 6. A Closing Note—The Pandemic’s Impact on Research

Finally, it is pertinent to bring into the discussion the tragic catastrophe that has struck the world and all its continents: COVID-19. The scientific community has played a vital role in many aspects for understanding this previously unknown virus and presenting methods for the treatment. But the effect on the scientific community has been terrible, with many laboratories shut down mainly due to concern of contact between infected and non-infected personnel. Fortunately, with the vaccines, we see a diminution of the closure of laboratories.

Saporin, as are many of the Type I ribosome-inactivating proteins, is an antiviral protein [181,182,183]. The effect of COVID-19 on research utilizing Saporin or its conjugates has been typical of the effect on all areas of the world economy: sometimes minimal, but also sometimes shutting down projects for months. As a commercial provider of Saporin products, Advanced Targeting Systems has seen a short, strong negative impact, followed by a return to a degree of normalcy. Then came the variants (Delta, Omicron).

We express our gratitude to those in in the world who are providing the mechanism for understanding the virus, for deriving vaccines against viral infection, for creating infection-detecting methods, and the many other contributions that help us to cope with it all.

We are all in this together.

## Figures and Tables

**Figure 1 toxins-14-00184-f001:**
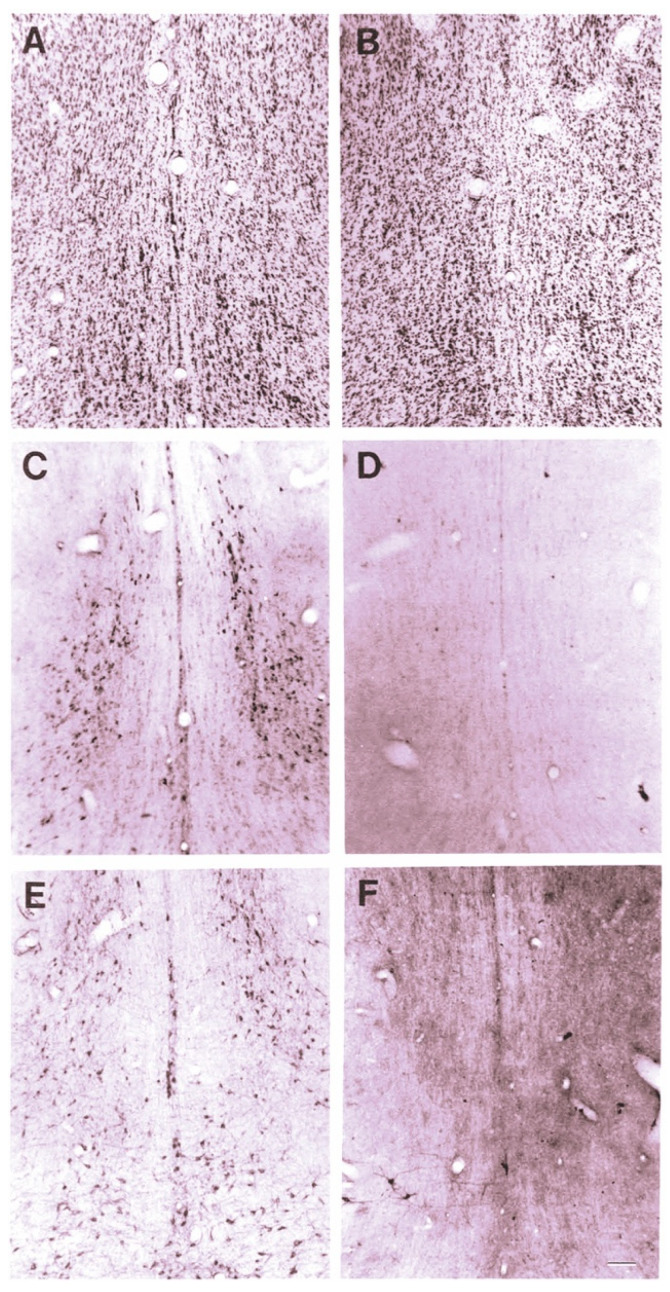
Cresyl violet staining of the medial septal region after ICV injection of 192-IgG-SAP. (**A**) normal superior cervical ganglia (SCG) section from a rat 5 days after injection (80 μg). (**B**) section from a rat 14 days after injection into the lateral ventricle (4 μg). (**C**) immunoperoxidase (ip) stain for ChAT from same (normal) rat as (**A**). (**D**) ip staining for ChAT from same rat as (**B**). (**E**) ip stain for p75 receptor from same (normal) rat as (**A**). (**F**) ip staining for p75 receptor from same rat as (**B**,**D**), showing near complete loss of positively-stained neurons. Bar in F = 100 μm and applies to all panels.

**Figure 2 toxins-14-00184-f002:**
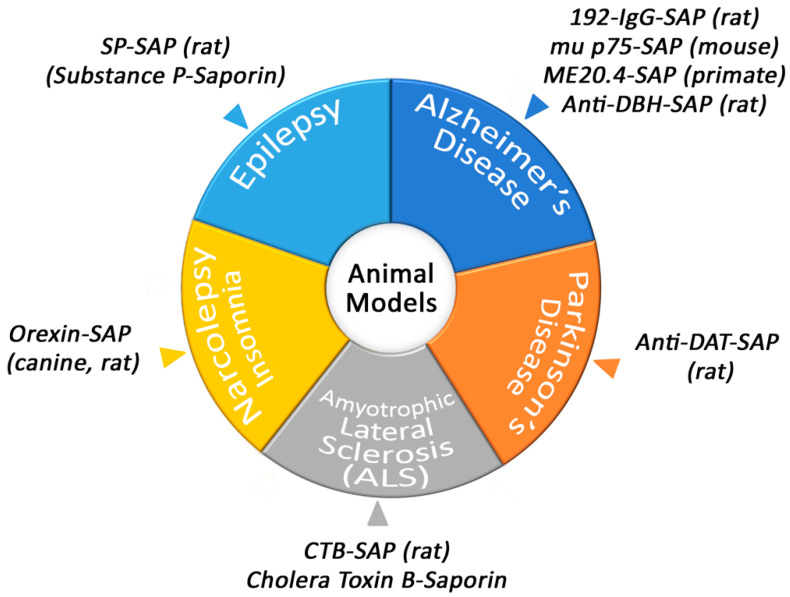
Animal models produced by Saporin conjugates.

**Figure 3 toxins-14-00184-f003:**
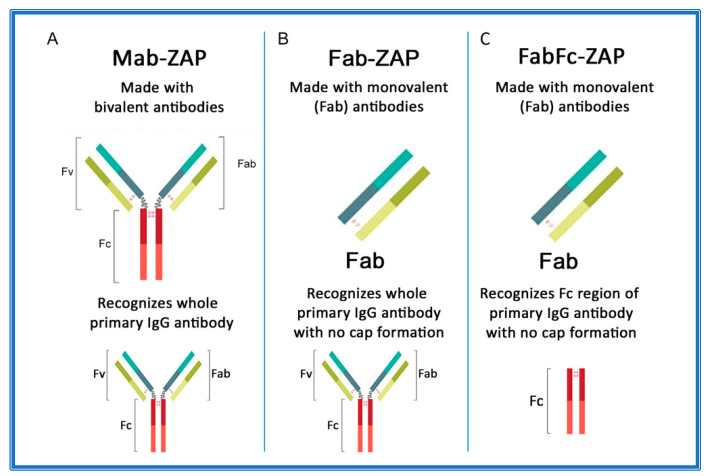
Antibody composition of Saporin secondary conjugates. (**A**) Mab-ZAP uses a bivalent antibody, consisting of both the fragment antigen-binding (Fab) region and the fragment crystallizable (Fc) region of IgG, and is capable of reacting with whole IgG. (**B**) Fab-ZAP uses a monovalent antibody to prevent capping and recognizes whole IgG. (**C**) FabFc-ZAP uses a monovalent antibody to prevent capping and recognizes the Fc region.

**Figure 4 toxins-14-00184-f004:**
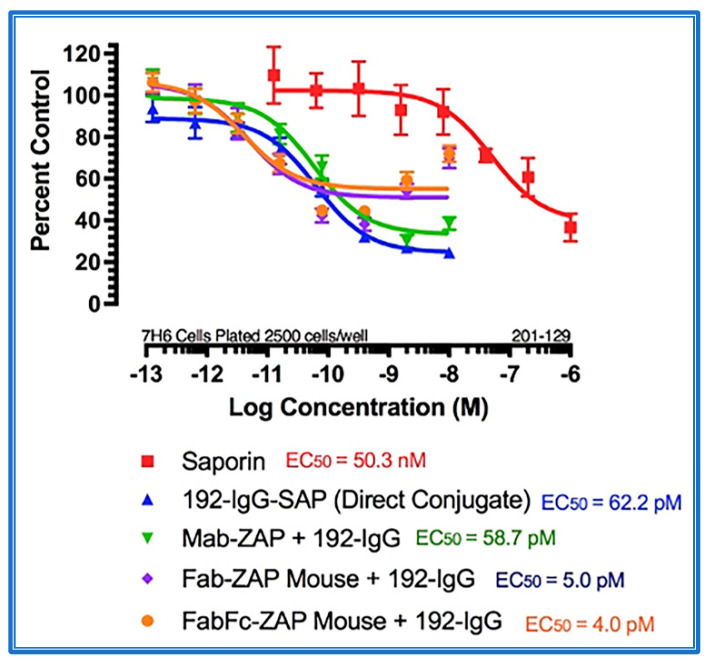
Cytotoxicity assay of bivalent and monovalent IgG-Saporin secondary conjugates. 7H6 cells, a clone of the rat C6 glioma cell line, were plated at 2500 cells/well. The secondary conjugates Mab-ZAP, Fab-ZAP mouse, and FabFc-ZAP mouse were reacted with monovalent 192-IgG as the targeting agent. A stoichiometric effect was seen when using Fab-ZAP (purple-diamond line) and FabFc-ZAP (orange-circle line) held at a constant concentration (4.5 nM), reacted with the titrated mouse monoclonal 192-IgG.

**Figure 5 toxins-14-00184-f005:**
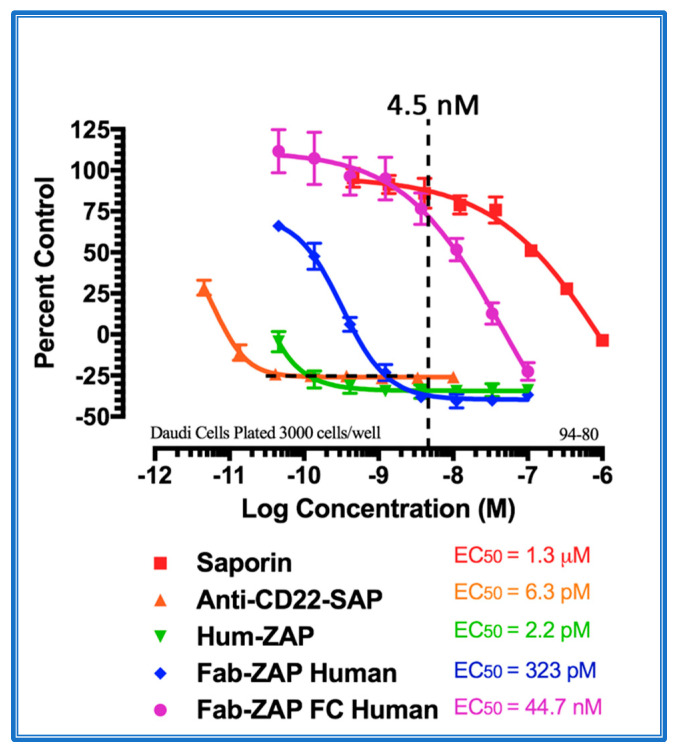
Cytotoxicity assay of Fc region-specific IgG-Saporin secondary conjugates. Daudi cells were plated at 3000 cells/well. Saporin alone and a direct conjugate of an antibody to CD22 and Saporin were used as controls. Three secondary conjugates were compared: (1) Hum-ZAP alone; whole IgG recognizing human whole IgG, (2) Fab-ZAP Human; monovalent IgG recognizing whole Human IgG, and (3) FabFc-ZAP Human; monovalent IgG recognizing only Human Fc. Data show that the two secondary conjugates that recognize whole IgG caused cell death without a primary antibody as a targeting agent due to endogenous sIg on the cell surface. Only the Fc-specific conjugate displayed cell death similar to the Saporin-negative control.

**Figure 6 toxins-14-00184-f006:**
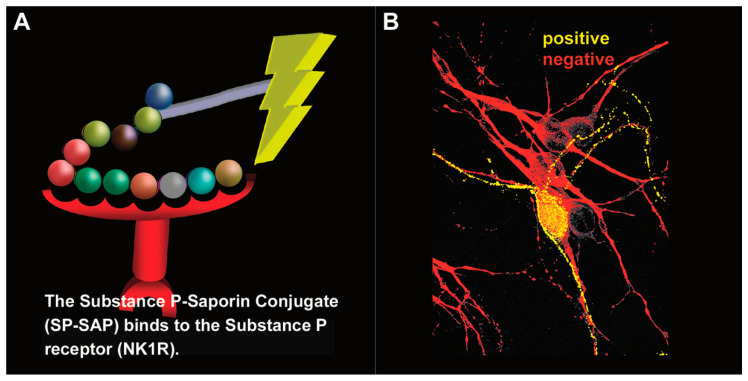
Substance P–Saporin (SP-SAP): composition and specificity. (**A**) SP-SAP is a conjugate between the peptide, Substance P, and the ribosome-inactivating protein, Saporin. This conjugate specifically targets neurons that express the substance P receptor (NK1R). (**B**) Saporin immunofluorescence is in yellow, showing specific targeting and internalization into a neuron that expresses NK1R. Neurons that do not express NK1R are shown in red.

**Figure 7 toxins-14-00184-f007:**
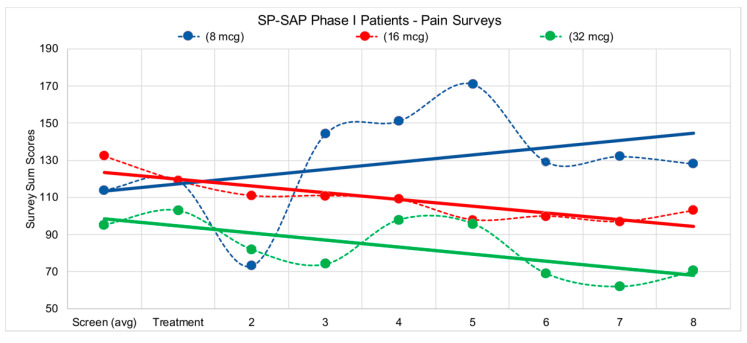
Patients in the study filled out several pain surveys, VAS (Visual Analog Scale) Pain, VAS Bothersome, ODI (Oswestry Disability Index), and BDI (Beck Depression Inventory). Survey scores were tabulated according to standard methods for each survey type and values were reported to sponsor. Surveys were administered first during the screening process, 1 week prior to treatment to initiate a baseline for patient pain levels, then again on day of treatment, followed by weekly surveys during each patient visit with the clinical staff following treatment. Due to the complex pain profile for eligible patients, trend lines were incorporated into data above to visualize the overall pain picture for each of the corresponding doses. Both the 16-mcg (microgram) patient and the 32-mcg patient reported reductions in pain over the 8 weeks following treatment.

**Table 1 toxins-14-00184-t001:** Varied uses of CTB-SAP to lesion motoneurons.

Area of Injection	Lesioned Cells	Reference
Intrapleural	Respiratory Motoneurons	[101,102,103,104,105]
Intralingual	Hypoglossal Motoneurons	[93,106]
Celiac Ganglia	Mesenteric Projecting Sympathetic Neurons	[107]
Bilateral Stellate Ganglia	Cardiac Sympathetic Neurons	[107,108,109]
Gastrocnemius Muscle	Spinal Motoneurons	[96]
Bulbocavernosus Muscle	Spinal Nucleus of the Bulbocavernosus Motoneurons	[94,95]
Vastus Medialis Muscle	Quadricep Motoneurons	[110]

**Table 2 toxins-14-00184-t002:** Molecules for targeting and internalization of Saporin.

Targeting Agent	Size
Antibody: Whole IgG	160 kDa
Antibody: F(ab’)2	110 kDa
Antibody: F(ab)	55 kDa
Antibody: single-chain variable fragment (scFv)	28 kDa
Lectin (e.g., Isolectin B4)	28 kDa
Growth Factor (e.g., Fibroblast Growth Factor (FGF)	16.5 kDa
RNA Aptamers	13–17 kDa
Peptides (e.g., Epidermal Growth Factor (EGF)	2–6 kDa
Extracellular vesicles (EVs)	
Quantum Dots	

**Table 3 toxins-14-00184-t003:** Representative non-pain related research with SSP-SAP.

Year	Application	Citation
2001	Focal inhibitory interneuron loss and principal cell hyperexcitability in the rat hippocampus after microinjection of a neurotoxic conjugate of Saporin and a peptidase-resistant analog of Substance P.	[160]
2002	Depressor and tachypneic responses to chemical stimulation of the ventral respiratory group are reduced by ablation of neurokinin-1 receptor-expressing neurons.	[161]
2002	Identification of a potential ejaculation generator in the spinal cord.	[162]
2003	A group of glutamatergic interneurons expressing high levels of both neurokinin-1 receptors and somatostatin identifies the region of the pre-Bötzinger complex.	[163]
2005	Elimination of rat spinal neurons expressing neurokinin 1 receptors reduces bladder overactivity and spinal c-fos expression induced by bladder irritation.	[164]
2007	From anxiety to autism: spectrum of abnormal social behaviors modeled by progressive disruption of inhibitory neuronal function in the basolateral amygdala in Wistar rats.	[165]
2008	Selective lesion of retrotrapezoid Phox2b-expressing neurons raises the apnoeic threshold in rats.	[166]
2008	Utilization of the least shrew as a rapid and selective screening model for the antiemetic potential and brain penetration of substance P and NK1 receptor antagonists.	[167]
2009	The neurokinin-1 receptor modulates the methamphetamine-induced striatal apoptosis and nitric oxide formation in mice.	[168]
2009	Anxiety-like behavior is modulated by a discrete subpopulation of interneurons in the basolateral amygdala.	[169]
2010	Transplant of GABAergic precursors restores hippocampal inhibitory function in a mouse model of seizure susceptibility.	[170]
2011	Ventilatory effects of Substance P-Saporin lesions in the nucleus tractus solitarius of chronically hypoxic rats.	[171]
2012	C1 neurons excite locus coeruleus and A5 noradrenergic neurons along with sympathetic outflow in rats.	[172]
2014	NK1-receptor-expressing paraventricular nucleus neurons modulate daily variation in heart rate and stress-induced changes in heart rate variability.	[173]
2014	Expression of different neurokinin-1 receptor (NK1R) isoforms in glioblastoma multiforme: potential implications for targeted therapy.	[174]
2017	Chemosensitive Phox2b-expressing neurons are crucial for hypercapnic ventilatory response in the nucleus tractus solitarius.	[175]
2019	contribution of the retrotrapezoid nucleus and carotid bodies to hypercapnia- and hypoxia-induced arousal from sleep.	[176]
2019	Episodic stimulation of central chemoreflex elicits long-term breathing disorders and autonomic imbalance in heart failure rats.	[177]
2019	Targeted hippocampal GABA neuron ablation by Stable Substance P-Saporin causes hippocampal sclerosis and chronic epilepsy in rats. (A new, stable model of temporal lobe epilepsy.)	[117]
2019	Spinal neuropeptide Y1 receptor-expressing neurons form an essential excitatory pathway for mechanical itch.	[178]
2020	A role for neurokinin-1 receptor expressing neurons in the paratrigeminal nucleus in bradykinin-evoked cough in guinea-pigs.	[179]
2021	Possible contribution of cerebellar disinhibition in epilepsy.	[180]

## Data Availability

Reference citations are included in this manuscript that show where the data can be found.

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
