# Peer review of "Saporin as a Commercial Reagent: Its Uses and Unexpected Impacts in the Biological Sciences—Tools from the Plant Kingdom"

_toxins, 2022, doi:10.3390/toxins14030184_

Round 1

Reviewer 1 Report

-The authors review the toxin saporin from Saponaria officinalis beyond the immunotoxin/cancer field. They present a cogent and robust review of saporin and its uses in Alzheimers, pain management, and other important applications. Generally, the material is presented in historical context render it very interesting to the readers of toxins. For example, its use in studies of nervous system biology.  It is well presented and will be of interest to the readers of toxins. However, there are shortcomings.

-My biggest disappointment is that there is no section outlining the advantages and disadvantages in cancer research. Cancer research was really the first frontier for the toxin and it arguably failed there. There should be a section explaining  as to why saporin has never reached phase 1 clinical studies for cancer. It would be best to feature this in a separate section. In other words, please present a succinct and thoughtful list of the impediments to saporin/antibody therapy. The authors mention that many of the problems have been resolved, but do not explain what problems exactly been resolved.

-Issues that could be addressed in this section include variability of potency in conjugates, non-target activity in vivo, controlling stoichiometry, tumor penetration, manufacturing costs, and the fact that saporin cannot be genetically engineered.

-Immunogenicity is a major concern of these drugs. Immunogenicity is mentioned, but the way it is written, gives the impression that the problem has been solved by Pastan et al. As mentioned, Pastan’s group has focused on PE toxin. Are the authors suggesting that epitope mapping would be useful to solve problems with saporin immunogenicity, the way it did for PE?

If one considers Pastan’s work a success story, then how does the Saporin story compare.  In fact, although immunogenicity is reduced by Pastan, it is not eliminated. (Also, it is not necessary to congratulate the Pastan group on line 61. It might be more prudent to mention that mox is one of the few targeted toxins that have achieved FDA approval for licensure for clinical use,  a remarkable accomplishment.

-Along these lines, why does the authors think that most targeted toxins have had clinical success in liquid versus solid tumors?

-As mentioned, the drug has not done as well in cancer research, then comparatively, why do think it will be a success in the neurotropic arena?

-The sentences describing Wiley’s interest in injection of FGF-SAP into the left corpus callosum on line 83  is confusing.  In fact, what was the importance of Wiley’s thoughts at the time?

-What’s the significance of SSP-SAP versus P-SAP, is it just a more stable derivative that negates the importance of P-SAP?

Reviewer 2 Report

In this manuscript, the authors review the use of saporin for research and drug development. Saporin is a ribosome inactivating protein (RIP) obtained from the seeds of Saponaria officinalis (common soapwort). RIPs are a group of proteins with rRNA N-glycosylase activity (EC 3.2.2.22) that catalyze the elimination of a specific adenine located in the sarcin-ricin loop (SRL) inactivating ribosomes, leading to irreversible inhibition of protein synthesis. Saporin, along with ricin and PAP, is one of the most widely used RIPs in research and the construction of immunotoxins that are used in cancer therapy. The authors review applications of saporin other than its use as part of antitumor immunotoxins, addressing very diverse aspects (Alzheimer’s Disease, narcolepsy, insomnia, amyotrophic lateral sclerosis, Parkinson’s Disease, itch, …).

General comments:

The authors provide information that is interesting to the readers of Toxins and the manuscript is written in an appropriate and scientific way. English language, grammar, punctuation, spelling, and general style are fine. On this basis, the document deserves to be published with minor changes.

Specific comments:

Page 1, line 25; page 4, line 146; page 7, line 280; page 10, line 404; page 13 line 525: it is not usual to put the name of the authors in the headings of the different sections, but if there is any reason for it, it is OK for me.

Page 1, line 29: it would be useful for readers to explain the enzymatic activity of the RIPs. They are rRNA N-glycosylases (EC 3.2.2.22), and in addition most of them possess adenine polynucleotide glycosylase activity (e.g., saporins).

Page 2, line 59: correct the phrase “his has been by …”

Page 5, line 206: change “icv” by “ICV”

Page 5, line 211: Is "Bar in F = 100 pm" correct?

Page 6, line 222: change “5,7-DHT” by “5,7-Dihydroxytryptamine (5,7-DHT)”

Page 7, line 282: “pharmaceutical research arenas” o “pharmaceutical research areas”?

Page 8, line 331: table 1 is not cited in the text.

Page 9, line 344: why has not itch been included in Figure 2? On the other hand, what kind of conjugate is IB4-SAP?

Page 9, line 371: are Anti-DBH-SAP and Anti-DβH-SAP (in Figure 1) the same immunotoxin?

Page 11, lines 456-460: figure 3 is confusing; the meaning of the initials referring to the fragments must be described and these fragments must be indicated appropriately (Fv seems to indicate the Fab fragment).

Page 11, line 464: indicate what type of conjugate is FabFc-ZAP.

Page 12, figures 4 and 5: are the results shown in these figures published? If so, indicate the references.

Page 14, line 573: change “SP SAP” by “SP-SAP”

Page 15, line 595: define mcg.

Figures: the review deserves better quality figures (especially figures 4, 5, and 7).

Round 2

Reviewer 1 Report

I now find this review suitable for publication.

Author Response

Thank you for your time in reviewing our article.